# Direct Observation of COVID-19 Prevention Behaviors and Physical Activity in Public Open Spaces

**DOI:** 10.3390/ijerph19031335

**Published:** 2022-01-25

**Authors:** Richard R. Suminski, Gregory M. Dominick, Norman J. Wagner, Iva Obrusnikova

**Affiliations:** 1Center for Innovative Health Research, Department of Behavioral Health and Nutrition, University of Delaware, Newark, DE 19726, USA; 2Physical Activity Measurement and Evaluation Laboratory, Department of Behavioral Health and Nutrition, University of Delaware, Newark, DE 19726, USA; gdominic@udel.edu; 3Chemical & Biomolecular Engineering, University of Delaware, Newark, DE 19716, USA; wagnernj@udel.edu; 4HBS Health and Disability Concentration and the Health and Disability Laboratory, Department of Behavioral Health and Nutrition, University of Delaware, Newark, DE 19726, USA; obrusnik@udel.edu

**Keywords:** infectious disease, public health, lifestyle behaviors, measurement

## Abstract

Mask wearing and physical distancing are effective at preventing COVID-19 transmission. Little is known about the practice of these behaviors during physical activity (PA). In this longitudinal study, direct observation was used to describe COVID-19 prevention behaviors among physically active individuals. The Viral Transmission Scan (VT-Scan) was used to assess COVID-19 prevention behaviors of people standing, sitting, walking, jogging, and cycling in educational, retail, and residential areas. The VT-Scan was performed once per week over 22 weeks between 11:00 a.m. and 2:30 p.m. Information was manually extracted from videos collected during VT-Scans. A total of 4153 people were described, of which 71.2% were physically active, 80.0% were 18–30 years of age, 14.0% were non-white, 61.0% were female, and were 19.6% obese. Individuals not engaged in PA were less compliant with COVID-19 prevention behaviors than physically active people. Compliance differed by PA type, with walkers less compliant with COVID-19 prevention behaviors than joggers and cyclists. Among those physically active, non-compliance with COVID-19 prevention behaviors was higher in 18–30-year-olds, whites, and men. Engagement in COVID-19 prevention behaviors varies as a function of PA. Efforts to promote compliance with recommendations may benefit from tailored messaging, taking into account PA participation, PA type, and characteristics of physically active individuals.

## 1. Introduction

The COVID-19 pandemic is driven by an airborne pathogen (severe acute respiratory syndrome coronavirus 2 (SARS-CoV-2)) that spreads by inhaling or ingesting respiratory droplets and aerosols expelled into the air from an infected human carrier. The virus is highly contagious and a recent mutation (Delta) is roughly two times more contagious than the original coronavirus strain [1]. Moreover, carriers can be both symptomatic and asymptomatic [2,3,4]. The burden COVID-19 has imposed on health systems, economies, and people throughout the world is unprecedented. Since the start of the COVID-19 pandemic in the beginning of 2020, 245 million cases with 5 million virus-related deaths have occurred globally, with 45.5 million confirmed cases and over 730,000 deaths reported in the United States alone [5].

Most governments have adopted various non-pharmaceutical interventions (i.e., COVID-19 prevention behaviors) to mitigate the spread of the virus with and without attaining desired vaccination levels and readily available treatments. These include social or physical distancing (staying at least six feet away from people not from your household in both indoor and outdoor spaces), the use of masks, and avoiding touching areas around the face (e.g., mouth, nose, or eyes) after contact with a fomite (object/material that could harbor infectious agents) [6]. Most evidence indicates that these behaviors are effective at diminishing the spread of SARS-CoV-2, including recently found variants [7,8,9].

Personal (e.g., racial background, weight status) and environmental (e.g., temperature) characteristics appear to enhance/inhibit the practice of COVID-19 prevention behaviors, and thus, their effectiveness [10,11]. For instance, women, elderly, non-white, and residents of large, urban areas may have certain perceptions and/or resources that could influence the likelihood of wearing a mask [10,12,13,14,15,16]. There is sparse evidence, however, concerning the practice of COVID-19 prevention behaviors by physically active individuals, particularly in outdoor public open spaces, where engagement in physical activity has increased since the onset of the pandemic [17]. Only one study could be located that provided information implying that study subjects were mobile or potentially physically active. In this study, 75.6% of pedestrians observed correctly wore a mask [18]. This gap in knowledge warrants attention for the following reason. Person-to-person, COVID-19 transmission primarily results from the release of infectious microorganisms into the air by expiratory activities including breathing [19]. More forceful exhalations, such as those associated with higher intensity physical activity, produce high-velocity droplet sprays that cause even larger quantities of infectious aerosol particles to be expelled into the environment [20]. Thus, masking and physical distancing would take on added importance during higher intensity physical activity performed in the presence of others. However, physically active individuals might be more apt to avoid engaging in COVID-19 prevention behaviors, as they may believe masks will limit their exercise performance, cause feelings of claustrophobia and/or impact breathing, especially during higher physical activity intensities when masking would be most desired [21].

Given that both physical activity and COVID-19 prevention behaviors are critical contributors to public health, particularly during the current pandemic, efforts are justified that promote maximum engagement in both. Research describing the interaction between these behaviors is necessary to inform the development of such efforts. Therefore, the purpose of the current study was to describe mask wearing and physical distancing behaviors in relation to physical activity performed in public open spaces using the direct observation method. For context, this study was conducted in an area where a State of Emergency was in effect that mandated mask wearing and physical distancing in indoor and outdoor areas with any failure to comply constituting a criminal offence.

## 2. Materials and Methods

### 2.1. Setting (Spatial and Temporal)

This longitudinal study was conducted in Newark, Delaware along a 7.5 km, pre-determined observation route that coursed through residential (35.4% of route), business (33% of route), and educational (31.6% of route) areas. Newark is a small town with a population of 33,448. According to the 2015 U.S. Census and national survey data corresponding to the study area, 56.9% of residents were female, 83.2% were white, 78.1% were 18–34 years of age, and 20.8% were obese [22]. The observation route was constructed to connect 79 nodes, which represented areas normally associated with high volumes of human traffic (crosswalks, entrances/exits of buildings, and public open spaces). The route between nodes consisted of sidewalks, streets, and walkways. Data were collected between the hours of 11:00 a.m. and 2:30 p.m. (heaviest human traffic volume times for the study area), once on a weekday, for 6, 8, and 8 weeks during fall, winter, and spring data collection periods, respectively. The first data collection day (e.g., Monday) of a data collection period was randomly selected, with the remaining data collection days occurring in a repeating sequence. No precipitation fell on any of the observation days; temperature (°C) of 18.1 +/− 8.7, 0.6 to 31.7; wind speed (kph) of 16.6 +/− 6.4, 4.8 to 30.0; relative humidity (%) of 49.7 +/− 18.7, 19.0 to 49.7; and barometric pressure (mm) of 1.18 +/− 0.01, 1.16 to 1.20 were typical for the area during the times data were collected [23].

### 2.2. Observation Procedure

The Viral Transmission-Scan (VT-Scan) was used to observe behaviors and characteristics of interest. Previous work demonstrated that the method displays good to excellent intra- and inter-reliability with intra-class correlation coefficients ranging from 0.836 to 0.997 [24]. This observation method was selected because it provides the most reliable, objective measures of place-based PA [25]. The method required a trained, mobile observer (one study investigator) to traverse the observation route while recording videos with a wearable video device. The wearable video device used was the Gogloo E7 SMART (Engine (HK) Co., Limited, ShenZhen, China), a state-of-the-art wearable video device indistinguishable from a pair of normal sunglasses. Videos were obtained for 10 s at each node, from the same position, on each observation day with the observer motionless. Videos were also captured continuously when the observer moved along walkways/sidewalks/streets from one node to another.

Information was manually extracted from the videos following VT-Scan procedures. Briefly, two trained reviewers used computer-aided manipulations (i.e., zooming, slow-motion, rewind, pause) and the VT-Scan rules to determine COVID-19 transmission behaviors related to face mask compliance (have a mask, have mask but wearing it incorrectly, no mask or have a mask but wearing it incorrectly), physical distancing compliance (<6 ft vs. >6 ft from other people), activity behaviors (not physically active = people sitting, standing, or prone and physically active = people not sitting, standing, or prone), and personal characteristics of sex/gender (male, female), age group (<18, 19–30, 31 to 55, and >55 years), race (white, non-white), and weight status (not obese, obese). The observed COVID-19 prevention behaviors were utilized to derive a “recommendation compliance” variable which combined mask and physical distancing (mask and physical distancing non-compliant versus other combinations) (6). Timestamps (h:min:s) shown on the videos and longitude and latitude coordinates (from Google Maps location tool) were used to mark when and where an individual was described.

### 2.3. Data Management

A protocol for handling the videos was developed to mitigate ethical issues arising from collecting, storing and, analyzing detailed video data. After each data collection period, the wearable video device’s micro secure digital card with acquired videos was removed and immediately transferred to a lock box for storage until transported to the computer lab. At the lab, the video data on the secure digital card was uploaded to a secure server and encrypted after which the secure digital card’s video files were erased. The encrypted video files were then distributed to reviewers who performed the reviews. Once reviews were complete, all videos in the possession of the reviewers were erased. The University’s Institutional Review Board for Protection of Human Subjects Committee waived the requirement for written informed consent for participants in this study under exemption #2: research involving the observation of public behavior, in accordance with the national legislation and the institutional requirements.

### 2.4. Statistical Analysis

Associations between meteorological conditions (temperature, humidity, wind speed, and barometric pressure) and COVID-19 prevention behaviors were examined using linear regression. A two-way analysis of variance (ANOVA) between groups was used to test for differences across the three seasons in the percentages of physically active and not physically active individuals who did not meet the recommendation each study week (*n* = 6, 8, and 8 weeks for the fall, winter, and spring, respectively) while controlling for temperature. Significant main effects for season and activity were examined in physically active and not physically active individuals using separate, one-way ANOVA procedures with temperature as a control variable. For season, pairwise comparisons of estimated marginal means (EMM) +/− standard errors (SE) and Bonferroni adjustments for multiple comparisons were conducted as a follow-up to the one-way ANOVAs to detect which seasons differed regarding non-compliance with the recommendation. Chi-square tests for independence with Yate’s continuity correction (for 2 × 2 tables only) were performed to determine if COVID-19 prevention behaviors varied by activity behaviors. The same Chi-square procedure was used to examine relationships between COVID-19 prevention behaviors and demographics in physically active individuals only. For the age group variable, individuals described as being under 18 years of age were removed from the analysis due to the small sample size and no one in this age group practicing physical distancing. All statistical analyses were performed using the Statistical Package for the Social Sciences software package with alpha set a priori at 0.05 [26].

## 3. Results

A total of 4153 people along the observation route were described over the course of 22 weeks with statistically equivalent (F(2, 21) = 1.7, *p* = 0.21) numbers described during the fall (M = 152.7, SD = 40.6 people), winter (M = 187.1, SD = 50.4 people), and spring (M = 217.5, SD = 89.1 people) data collection periods. Most were 18–30 years of age (80.0% with 0.8% under 18, 15.1% between 31 and 55, and 4.1% over 55), white (86.0%), female (61.0%), and not obese (80.4%). Also provided in Table 1 are the sample sizes for each characteristic. Characteristics for most people observed (*n* = 4153) were described (96.3% for age group (*n* = 3998 described); 95.6% for race (*n* = 3970); 99% for sex/gender (*n* = 4112); 95.1% for weight status (*n* = 3951)).

The majority of people described were engaged in physical activity (71.2%; 2956/4153) and most of the physically active people were walking (97.2%; 2872/2956). Standers/sitters accounted for 28.8%, joggers for 0.9%, cyclists for 0.8%, and skateboarders for 0.3%. Mask and physical distancing non-compliance were 54.6% and 53.6%, respectively, while non-compliance with the recommendation to wear a mask when within six feet of another person was 32.7%.

According to the linear regression analyses, only ambient temperature was significantly associated with COVID-19 prevention behaviors in physically active individuals. Mask non-compliance (t = 3.9, *p* = 0.001) and non-compliance with the recommendation (t = 2.7, *p* = 0.02) increased as temperatures increased. None of the meteorological variables were significantly associated with physical distancing.

Individuals not engaged in physical activity were less compliant with mask wearing (*X*^2^(1, N = 4015) = 200.4, *p* < 0.001), physical distancing (*X*^2^(1, N = 4151) = 71.6, *p* < 0.001), and the recommendation (*X*^2^(1, N = 4096) = 218.7, *p* < 0.001) than people engaged in physical activity (Table 1).

Shown in Figure 1 are the estimated marginal means (EMM) for the percent of physically active and not physically active (sitting/standing) people non-compliant with the recommendation for each of the three seasons. There were significant main effects for season (F(2, 37) = 7.7, *p* = 0.002, partial η^2^ = 0.29) and activity (F(1, 37) = 37.1, *p* < 0.001, partial η^2^ = 0.50). Among physically active individuals, non-compliance with the recommendation was significantly lower in the fall (EMM =11.9%, SE = 4.4%) compared to the winter (EMM = 30.0%, SE = 4.3%) (F(2, 18) = 4.3, *p* = 0.03, partial η^2^ = 0.32). Spring non-compliance with the recommendation (EMM = 28.2%, SE = 3.7%) did not differ significantly from fall (*p* = 0.36) or winter (*p* = 0.91) non-compliance rates. Non-compliance with the recommendation did not differ significantly between seasons in non-physically active individuals (F(2, 18) = 2.0, *p* = 0.17, partial η^2^ = 0.18). Physically active individuals displayed significantly lower rates of non-compliance with the recommendation than those not physically active in the fall (F(1, 9) = 16.1, *p* < 0.001, partial η^2^ = 0.62; EMM = 11.9.%, SE = 4.4% vs. 33.4%, SE = 4.4%), winter (F(1, 13) = 6.3, *p* = 0.02, partial η^2^ = 0.33; EMM = 30.0%, SE = 4.3% vs. 45.8%, SE = 4.3%), and spring (F(1, 13) = 13.7, *p* = 0.003, partial η^2^ = 0.51; EMM = 28.2% +/− SE = 3.7% vs. 46.8%, SE = 3.7%).

Mask non-compliance (*X*^2^(2, N = 2856) = 26.23, *p* < 0.001), physical distancing non-compliance (*X*^2^(2, N = 2941) = 38.9, *p* < 0.001) and recommendation non-compliance (*X*^2^(2, N = 2916) = 7.9, *p* = 0.019) varied by physical activity type. Joggers were least likely to be mask compliant whereas walkers were the least compliant with physical distancing and the recommendation (Table 2).

COVID-19 prevention behavior non-compliance rates among physically active individuals are shown in Table 3 by personal characteristics. Statistics for significant Chi-square results are provided in the table. Mask non-compliance was significantly higher in whites compared to non-whites, males versus females, and non-obese compared to obese. Rates of mask non-compliance were similar among the three age groups (*X*^2^(2, N = 2767) = 1.78, *p* = 0.41). Physical distancing non-compliance was significantly higher in the 18–30 age group as compared with the other two age groups and males versus females. Physical distancing non-compliance did not differ significantly between whites and non-whites (*X*^2^(1, N = 2822) = 1.75, *p* = 0.19) or non-obese and obese (*X*^2^(1, N = 2834) = 1.01, *p* = 0.32). Non-compliance with the recommendation was significantly higher in the 18 to 30 age group versus the other two age groups, whites compared to non-whites, and males compared to females. Non-obese and obese displayed similar rates of non-compliance with the recommendation (*X*^2^(1, N = 2821) = 0.33, *p* = 0.57).

## 4. Discussion

The purpose of the current study was to describe mask wearing and physical distancing behaviors in relation to physical activity performed in outdoor, public open spaces using the direct observation method. Longitudinal data, collected during different seasons, indicated that physically active versus non-physically active individuals are more compliant with COVID-19 prevention behaviors. Compliance also varied as a function of physical activity type with walkers less likely than joggers and cyclists to comply with the recommendation to wear a mask when within six feet of another person. Subgroups of physically active individuals displayed less compliance. More specifically, younger adults (18–30 years of age), whites, and males were less compliant.

Compliance with COVID-19 prevention behaviors is important for reducing the spread of the disease even with the availability of vaccines. The practice of these behaviors varies by personal and other characteristics [10,11]. The current study adds to this body of literature by showing mitigation behaviors also vary by activity participation with sitters and standers less likely than physically active to be compliant. Although reasons for compliance/non-compliance were not ascertained in this study, it is reasonable to assume that the prevention behaviors observed were partially related to current regulations. Most of the non-active were sitting at tables where table dimensions did not permit physical distancing. In addition, people at tables could remove masks according to local and State mandates under the guise that they were drinking or eating (even though tables were in public areas and not directly associated with a restaurant). This finding could be relevant for devising policies or interventions to increase compliance. For instance, point-of-decision prompts encouraging mask use and physical distancing could be strategically placed near sitting areas, similar to the placement of prompts encouraging stair versus escalator/elevator use [27]. This intervention type is easy to implement, low-cost, and can be rapidly altered to address seemingly continuously changing COVID-19 prevention policies.

To our knowledge, this is the first study to explicitly describe multiple COVID-19 prevention behaviors as practiced by physically active individuals in outdoor settings. Previous studies using observation described only mask wearing in “pedestrians” without mention of physical activity type [18,28,29]. The 54.6% non-compliant with mask wearing found in the current study is similar to mask non-compliance rates of people in outdoor settings reported by others (54.4%, 67.8%, and 40.2% for an average of 54.1%) [18,28,29]. None of these studies reported physical distancing alone or with mask use. Nevertheless, the observation method appears to be consistent in measuring mask compliance. The results of the present study also indicate that physical activity type is important to know when evaluating COVID-19 prevention behavior practices among individuals physically active in outdoor areas. Mask non-compliance was nearly two-fold higher in joggers than walkers and cyclists. This could be an area of concern if jogging were consistently performed at higher intensities resulting in more forceful exhalations. Physical distancing non-compliance was substantially higher in walkers compared with joggers and cyclists, who exhibited very low physical distancing non-compliance rates. This was probably the result of most joggers and cyclist being alone, which may reflect a common characteristic of such exercises [30]. On the other hand, a higher percentage of walkers compared with joggers and cyclists (data not shown) were observed along the business and educational portions of the observation route. Pedestrian density was higher in these areas compared with the residential areas. As such, a comparatively greater percentage of walkers would have found it more difficult to maintain a distance of six feet from other people. Given that 50.6% of walkers were not physical distancing and 26.3% were not compliant with the recommendation, walkers may be particularly vulnerable to contracting/spreading COVID-19. Although additional research is needed to confirm such a contention, information of this nature is valuable for messaging and recommendation formation about practicing proper COVID-19 prevention practices, particularly among potentially vulnerable populations.

COVID-19 prevention behaviors among the physically active were related to the season of the year and several personal characteristics, the latter being a common finding, albeit without reference to physical activity [10,11,12]. Non-compliance with the recommendation was lowest in the fall and highest in the spring. This pattern is particularly interesting, as it appears to coincide with external events occurring during the study. Although mandates requiring mask wearing and physical distancing outside and indoors were in effect during all three seasons, they were aggressively enforced with threat of criminal charges in the fall compared with the winter and spring. The results indicate that people physically active in outdoor settings are more apt than individuals not physically active in the same settings to regress towards non-conformity with COVID-19 mandates when permitted by relaxed enforcement. It is possible this finding is associated with a condition known as “mask fatigue”, defined as a “lack of energy that accompanies, and/or follows prolonged wearing of a mask” [31]. While mask fatigue can reduce compliance with mask wearing, it can also cause individuals to become physically and mentally tired due to the use of masks. Therefore, the physically active may be impacted more by mask fatigue than the non-physically active given the additional effects of the condition on physical performance. Physically active individuals not only become tired of wearing a mask, but they also “sense” that mask use is interfering with their physiological functions (e.g., breathing) during active periods, leading them to utilize masks even less. An interesting follow-up to this finding would be to examine efforts to alleviate the subjective “feelings” of mask fatigue and subsequent behavioral consequences.

Previous research has consistently found that COVID-19 non-compliance rates vary as a function of personal characteristics [10,11,12,16,18]. For instance, Rahimi et al. [18] reported rates of mask non-compliance of 61% in men and 40% in women, while Rader and colleagues [11] found 2.8-fold higher mask non-compliance rates in whites compared to blacks and Hispanics. Additionally, overweight or obese individuals tend to be more compliant with COVID-related behaviors than normal weight people [16]. Evidence from the current study, using direct observation, supports these findings. Physically active men, whites, and non-obese people were more likely to be mask non-compliant than physically active women, non-whites, and obese people, respectively. As stated above, determining why certain personal characteristics coincide with COVID-19 prevention behaviors was beyond the scope of the current study. However, a myriad of reasons have been offered to explain this connection, including perceived risk (e.g., whites perceive less risk of contracting and dying from COVID-19 than other racial groups, resulting in less compliance), social and economic factors (e.g., those with more resources to address illness may feel less pressure to conform), personal beliefs/attitudes (e.g., men view mask wearing as less masculine than women, which deters the behavior when in public), and feelings of self-worth (obese report feeling less well than non-obese, which is related to more social isolation) [10,12,13,14,15,16].

This study has strengths worth mentioning. Outcomes were assessed using an objective methodology that has advantages over more commonly employed methods such as self-report surveys [32]. Further, the observation method used was developed specifically for examining COVID-19 prevention behaviors, and it has been shown to have high intra- and inter-reliability. No other studies in this area achieved this level of measurement rigor. The study design was longitudinal, with data collected over 22 weeks. This provides more confidence in the outcomes as they would tend to be more representative of habitual behavior patterns than would results from a single, cross-sectional assessment. Relatedly, behaviors were observed during a wide range of meteorological conditions. According to the regression analysis, non-compliance with COVID-19 prevention behaviors among those physically active increased as temperature increased. Most likely, masks became more uncomfortable during higher temperatures; however, this contention is speculative and additional research is warranted on the connection between meteorological circumstances and COVID-19 prevention behaviors. Nonetheless, the finding that temperature was associated with COVID-19 prevention behaviors is consistent with previous research indicating that meteorological conditions influence outdoor behavior patterns [33].

There also are limitations to consider. First, as with many studies, caution is advised when generalizing to other areas dissimilar to where the current study was conducted—a small city on the east coast of the United States. However, it is possible the findings are relevant to a broad range of places as no evidence could be located suggesting other streetscape configurations/designs in residential, business, and educational districts would somehow alter behavior patterns and behaviors’ relationships with, for instance, personal characteristics. Nevertheless, a larger-scale study including a geographically heterogeneous sample is recommended to provide a more definitive picture of COVID-19 prevention behaviors and physical activity. The observation method used, while being comprehensive, only provides information that is “visible”. For instance, the nature of the relationships between individuals could not be discerned and this would affect recommendation compliance rates, because a mask is only required when within six feet of others not in your household or close group. Therefore, non-compliance rates are likely overestimated, but any overestimation would be expected to remain consistent across the subgroups (e.g., physical activity type) examined, and thus, not liable to significantly alter comparative analyses outcomes. Likewise, personal characteristics are inferred, not measured (e.g., weight status) and may be subject to a certain degree of error. However, VT-Scan does employ strict rules for ascertaining personal characteristics and it is highly reliable. Further, personal characteristic prevalence rates observed in the study were very similar to those reported for the residents of the study area (19.6% observed obese vs. 20.8% reported obese for the study area) [22]. Still, validation metrics for VT-Scan would add confidence in the method.

## 5. Conclusions

This study indicates that compliance rates with COVID-19 prevention behaviors vary as a function of physical activity. Future investigations are recommended to elucidate reasons compliance rates differ between people physically and not physically active in outdoor settings as well as between people performing different types of physical activity in outdoor spaces. In addition, it would be beneficial to examine the possible moderator/mediator roles personal characteristics may exert on the relation between physical activity and COVID-19 prevention behaviors. For example, recommendations and interventions could be informed by research looking at how one’s perceived risk of contracting COVID-19 impacts variations in prevention behaviors (and the subsequent probability of acquiring the disease) among physically active individuals performing different physical activities. At this point, it may also be relevant for researchers and practitioners to devise solutions to lowering non-compliance rates. Whether solutions involve local or state policy implementation, and/or some form of intervention (e.g., signage, education), their effectiveness might be enhanced if tailored based on physical activity (participation, type) as well as certain characteristics of individuals who are physically active in outdoor spaces.

## Figures and Tables

**Figure 1 ijerph-19-01335-f001:**
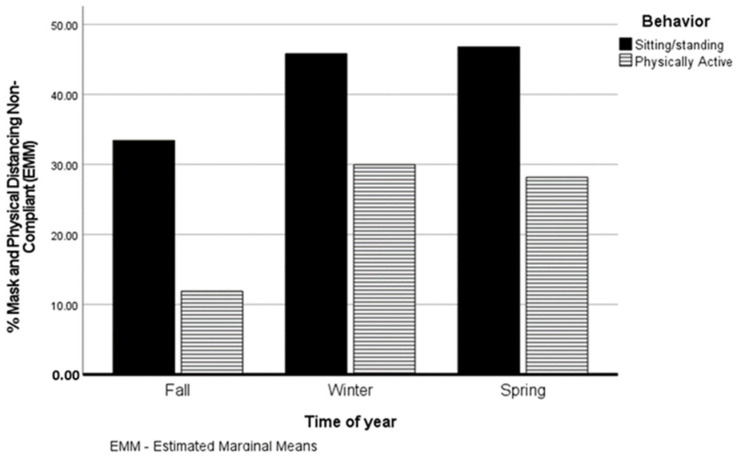
Percentages of non-physically active and physically active individuals per study week non-compliant with the recommendation for each of the three seasons.

**Table 1 ijerph-19-01335-t001:** COVID-19 behaviors in non-physically active and physically active individuals.

COVID-19 Behavior	Not Physically Active	Physically Active ^a^
Mask non-compliant	72.9 *	54.6
Physical distancing non-compliant	64.0 *	49.4
Recommendation non-compliant	49.9 *	25.8

* Not physically active different than physically active *p* < 0.001; ^a^ comprised of walkers, bikers, cyclists, and skateboarders.

**Table 2 ijerph-19-01335-t002:** COVID-19 behaviors by physical activity type.

COVID-19 Behavior	Walking	Jogging	Cycling
Mask non-compliant	46.9	89.2 ^a^	48.4
Physical distancing non-compliant	50.6 ^b^	13.5	12.1
Recommendation non-compliant	26.3 ^c^	10.8	12.1

^a^ Joggers > walkers and cyclists, *p* < 0.001; ^b^ walkers > joggers and cyclists, *p* < 0.001; ^c^ walkers > joggers and cyclists, *p* < 0.05.

**Table 3 ijerph-19-01335-t003:** COVID-19 behaviors among physically active individuals by personal characteristics.

COVID-19 Behavior	Mask Non-Compliant	Physical Distancing Non-Compliant	Recommendation Non-Compliant
18–30	49.1	51.4 ^d^	28.0 ^f^
31–55	45.6	43.9	19.3
55+	47.3	38.9	22.3
White	50.9 ^a^	50.3	27.9 ^g^
Non-white	35.0	46.7	19.2
Male	50.5 ^b^	41.3 ^e^	23.8 ^h^
Female	45.9	55.1	27.5
Not obese	49.7 ^c^	48.6	26.7
Obese	43.4	51.1	25.4

^a^ (*X*^2^(1, N = 2767) = 34.76; *p* < 0.001); ^b^ (*X*^2^(1, N = 2851) = 5.51; *p* = 0.019); ^c^ (*X*^2^(1, N = 2771) = 6.69; *p* = 0.01); ^d^ (*X*^2^(2, N = 2813) = 13.48; *p* < 0.001); ^e^ (*X*^2^(1, N = 2930) = 53.18; *p* < 0.001); ^f^ (*X*^2^(2, N = 2799) = 14.66; *p* < 0.001); ^g^ (*X*^2^(1, N = 2807) = 13.07; *p* < 0.001); ^h^ (*X*^2^(1, N = 2909) = 4.94; *p* = 0.026).

## Data Availability

The data presented in this study are available on request from the corresponding author. The data are not publicly available due to privacy concerns.

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
