# Peer review of "Direct Observation of COVID-19 Prevention Behaviors and Physical Activity in Public Open Spaces"

_ijerph, 2022, doi:10.3390/ijerph19031335_

Round 1
Reviewer 1 Report
In my opinion, the problem described in the article by the authors is very topical and very important. On the other hand, the data collected in the described manner and their subsequent analyzes are more suitable for publication as popular science curiosities, and not as a scientific article.
The research methodology raises serious doubts. In my opinion, it is not reliable. For example, qualifying on the basis of observations to the group of active or inactive people, to the group of obese or not people, and even more so to determine the age ranges, is doubtful as to the accuracy.
In addition, the description of all parts of the article, in my opinion, is flawed by many errors, and the purpose and conclusions are of very dubious quality.
Based on my experience, I cannot accept such a scientific work. Due to the respect for the authors' work, I am asking the reviewer to appoint other reviewers who may have different experiences in publishing such data.
Reviewer 2 Report
The article concerns an important and current research problem and I assess it positively. She recommends indicating a few additions:
1. A research gap should be identified.
2. Application forms should be more detailed.
3. The directions of further research should be more precisely specified.
Reviewer 3 Report
Dear Authors,
Thank you for your manuscript. The topic is important and relevant to the current situation of the COVID-19 pandemic. The paper is well written and well-structured. Below are my comments.
In the Introduction section, I suggest adding a detailed description of governmental regulations and restrictions during the study period as they might have an impact on subjects behavior. For example, wearing masks outside in some countries were and still are not obligatory. On contrary, in some counties during the most challenging periods of pandemic even going outside without an important reason was restricted and fined. So explaining the context of governmental regulations when analysing human behavior during the pandemic would be very helpful.
Minor comments. Please separate a subsection of statistical analysis. Also, I suggest reducing the number of tables by removing Tables 1 and 2 as they contain only a few numbers and providing this information in the text. Statistical comparisons and the p-values should be added to Table 4.
Also, I have some doubts according to the statistical techniques used. For example, why comparison of the percentages of behaviors during different seasons was performed with ANOVA? Why did the authors treat the percentage as continuous data? Also, it seems that the figure on page 6 does not have a title.
Round 2
Reviewer 3 Report
Dear Authors,
Thank you for taking into account my comments and suggestions.